# BUILDING BRIDGES, NOT WALLS: ADVANCING INTERPRETABILITY BY UNIFYING FEATURE, DATA, AND MODEL COMPONENT ATTRIBUTION

## ABSTRACT

The increasing complexity of AI systems has made understanding their behavior and building trust in them a critical challenge, especially for large language models. Numerous methods have been developed to attribute model behavior to three key aspects: input features, training data, and internal model components. However, these attribution methods are studied and applied rather independently, resulting in a fragmented landscape of approaches and terminology. We argues that feature, data, and component attribution methods share fundamental similarities, and bridging them can benefit interpretability research. We conduct a detailed analysis of successful methods of these three attribution aspects and present a unified view to demonstrate that they employ similar approaches: perturbations, gradients, and linear approximations. Our unified view enhances understanding of attribution methods and highlights new directions for interpretability and broader AI areas, including model editing, steering, and regulation.

## 1 INTRODUCTION

As AI systems grow increasingly complex, understanding their behavior and building trust in them remains a critical challenge, especially for large language models (LLMs) (Arrieta et al., 2020; Longo et al., 2024). Researchers have developed methods to explain AI systems by attributing their behavior to three distinct aspects: input features, training data, and internal model components. Feature attribution methods identify influence of input features at test time, revealing which aspects of the input drive the model's output (Zeiler & Fergus, 2014; Ribeiro et al., 2016; Horel & Giesecke, 2020; 2022; Lundberg & Lee, 2017; Smilkov et al., 2017). Data attribution analyzes how training data shape model behavior during the training phase (Koh & Liang, 2017; Ghorbani & Zou, 2019; Ilyas et al., 2022). Component attribution examines the internal workings of the model by analyzing how specific components, such as neurons or layers in a neural network (NN), affect model behavior (Vig et al., 2020; Meng et al., 2022; Nanda, 2023; Shah et al., 2024). While numerous attribution methods have been developed for each of these three aspects, and some survey papers have been published (Guidotti et al., 2018; Covert et al., 2021; Wang et al., 2024; Hammoudeh & Lowd, 2024; Bereska & Gavves, 2024), they have been studied and used rather independently by different communities, creating a fragmented landscape of methods and terminology for similar ideas.

**Our position is that feature, data, and component attribution methods can be bridged to advance not only interpretability research, by stimulating cross-aspect knowledge transfer, but also broader AI research, including model editing, steering, and regulation.** We show that these three types of attribution employ common methods and they differ primarily in perspective rather than core techniques. In the following sections, we first formalize a unified attribution problem that encompasses all three aspects to show these seemingly distinct approaches fall under the same framework (§2). We then examine the evolution of each attribution aspect and analyze its successful methods, revealing how these methods are connected through shared techniques and concepts, including perturbations, gradients, and linear approximations (§3, §4, §5). Building on this analysis, we present a unified view and illustrate shared concepts (§6.1), identify common challenges (§6.2), and highlight how this unified perspective facilitates cross-aspect knowledge transfer for new research development in interpretability (§6.3), and broader AI research (§6.4). In summary, we believe that this unified view enhances our understanding of attribution methods, bridges the current

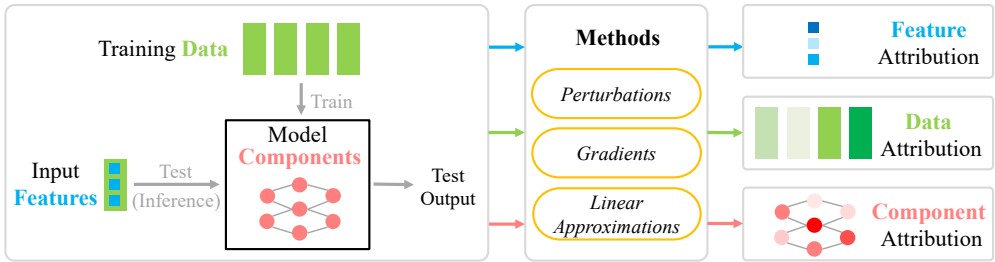

Figure 1: The three types of attribution regarding input features, training data, and internal model components.

fragmented landscape, makes the field more accessible to newcomers, and provides new insights and research directions.

## 2 THE ATTRIBUTION PROBLEM

Researchers have developed various attribution methods to analyze model behavior from different perspectives. We formally introduce three types of attribution problems and show they fall under a unified framework. Consider a learning problem with $d$-dimensional input features $x = [x_1, x_2, \ldots, x_d]$. During training, a dataset of $n$ data points: $\mathcal{D}_{\text{train}} = \{x^{(1)}, x^{(2)}, \ldots, x^{(n)}\}$ is used to train a model $f_\theta$ with parameters $\theta$ and components $c = \{c_1, c_2, \ldots, c_m\}$ by optimizing the loss function $\mathcal{L}(\theta)$. At test (inference) time, the model generates an output $f_\theta(x^{\text{test}})$ for a new input $x^{\text{test}}$. For notational simplicity, we omit $\theta$ and "test" and use $f$ and $x$ when the context is unambiguous. A notation summary is in Appendix A. The core objective of all three problems is to attribute the model's output $f(x)$ to different elements and quantify their influence with *attribution scores*.

*Feature attribution* quantifies how input features influence model outputs. These features may represent pixels in images, tokens in text, or other domain-specific units. We denote the attribution score of feature $x_i$ as $\phi_i(x)$.

*Data attribution* analyzes how training data shape model behavior. We quantify the influence of each training point $x^{(j)} \in \mathcal{D}_{\text{train}}$ through its attribution score $\psi_j(x)$.

*Component attribution* studies the role of model components in generating outputs. The components can have various definitions, such as neurons or layers in a NN. We denote the attribution score of component $c_k$ as $\gamma_k(x)$.

As illustrated in Figure 1, these three attribution problems share a fundamental connection: they all seek an *attribution function* $g$ that assigns scores to specific elements (features $x_i$, training points $x^{(j)}$, or components $c_k$) for a given test output $f(x)$, differing only in the choice of elements.

## 3 UNDERSTANDING FEATURE ATTRIBUTIONS

Feature attribution quantifies how individual features $x_i$ of an input $x$ influence a model's output $f(x)$ through attribution scores $\phi_i(x)$. Applied to model inference at test time, it explains model behavior without altering model parameters. The attribution results can be used to perform feature selection, identify spurious correlations, and justify model predictions to gain user trust. Feature attribution methods can be broadly classified into three categories: *perturbation-based methods*, *gradient-based methods*, and *linear approximation methods*. We discuss some prominent methods in each category below and provide more details in Appendix C.

### 3.1 PERTURBATION-BASED FEATURE ATTRIBUTION

Perturbation-based methods attribute feature importance by measuring how model outputs change when input features are modified and especially removed. They are also referred to as *removal-based methods* (Covert et al., 2021).

**Direct Perturbation** represents a straightforward application of perturbation analysis. The pioneering *Occlusion* method (Zeiler & Fergus, 2014) in computer vision replaces image pixels with grey squares and measures changes in the model's prediction. The method assumes that occluding crucial pixels will significantly impact the output. For images, pixel attribution scores create a *saliency map* highlighting the most influential regions. RISE (Petsiuk, 2018) advanced this approach by perturbing multiple image regions and combining their attribution results. The final attribution score weighs each attribution result by the model's predicted probability for that perturbed image.

**Game-Theoretic Perturbation** While intuitive, direct perturbation fails to capture synergistic interactions between multiple features. Cooperative game theory addresses this limitation by modeling features as players collaborating toward the model's output. The Shapley value (Shapley, 1953) provides a foundational solution within this framework and has inspired numerous feature attribution methods (Sundararajan & Najmi, 2020). Computing Shapley value attributions involves measuring a specific type of perturbation: how adding a feature $x_i$ to different feature subsets changes the model's output compared to the subset alone, known as the marginal contribution of $x_i$ to the subset. The final attribution score captures feature interactions by aggregating these marginal contributions across all possible feature subsets. Although theoretically sound, Shapley value methods face computational challenges as their complexity grows exponentially with feature dimensionality. To overcome this challenge, various approximation methods have been proposed, and Kernel SHAP (or simply SHAP) introduced by Lundberg & Lee (2017) has gained widespread adoption because of its efficient kernel-based approximation.

**Perturbation Mask Learning** is based on the idea that perturbation of including or excluding features can be viewed as applying a binary mask for each feature. Mask learning methods advance this idea by using learnable masks representing feature inclusion probabilities, which offer more nuanced control compared to binary masks. Dabkowski & Gal (2017) pioneered this approach for image classification by introducing a *masking model* that generates pixel masks, aiming to identify a minimal set of features that sufficiently maintain the prediction of the original input. The masking model acts as the attribution function $g$, where mask values represent feature attribution scores. While initial training is required, the masking model generates masks through a single forward pass at test time, which significantly improves runtime compared to earlier perturbation methods. For the mask learning methods, the main challenge is balancing feature minimality and predictive power.

### 3.2 GRADIENT-BASED FEATURE ATTRIBUTION

Gradients have emerged as a powerful tool for feature attribution. Gradients of model outputs $f(x)$ with respect to input features $x$, $\nabla_x f(x)$, quantify output sensitivity to small input changes (Erhan et al., 2009; Baehrens et al., 2010), measuring feature influence without requiring perturbations. Gradient-based attribution has superior computational efficiency compared to perturbation-based methods. While the latter requires $O(d)$ model evaluations for $d$ features, gradient-based approaches need only a single or a few forward and backward pass(es) to compute $\nabla_x f(x)$.

Gradient-based feature attribution emerged from computer vision, where it gained widespread adoption for generating attribution scores as image saliency maps (Simonyan et al., 2013), also known as *sensitivity maps* (Smilkov et al., 2017). The "vanilla gradients" method uses the gradients of the output class (log)probability with respect to input pixels as attribution scores (Simonyan et al., 2013). Since then, researchers have proposed numerous enhanced gradient-based methods. For example, Gradients $\times$ Input (Shrikumar et al., 2017) multiplies gradients with input values, Integrated Gradients (Sundararajan et al., 2017) accumulates gradients along a path from a baseline to the actual input, and Integrated Hessians (Janizek et al., 2021) further extends the analysis to feature interactions by computing the Hessian matrix. These methods leverage different gradient formulations to provide more accurate and stable attribution scores. A notable advancement is SmoothGrad (Smilkov et al., 2017), which generates multiple copies of the input with added Gaussian noise and computes sensitivity maps for each noisy sample. By averaging these maps, SmoothGrad reduces noise while preserving salient features that consistently influence model outputs.

### 3.3 LINEAR APPROXIMATION FOR FEATURE ATTRIBUTION

Linear approximation methods offer an alternative approach to feature attribution by fitting a simple linear surrogate model around the input of interest. These methods approximate the complex be-

havior of $f$ near a specific input $x$ using a linear model, normally in the form of $g(x) = w^\top x + b$ with coefficients $w$ and bias $b$. Then the coefficient $w_i$ directly provides a feature attribution score of feature $x_i$.

LIME (Ribeiro et al., 2016) exemplifies this approach. It samples instances around the input of interest, obtains model predictions for these samples, and fits a sparse linear model to capture the local model behavior. An innovation of LIME is its use of binary indicators (0 or 1) rather than actual feature values as inputs to the linear model, only representing feature inclusion or exclusion. The resulting linear model coefficients directly explain how each feature's presence influences the approximated model's output. Later, a variant of LIME called C-LIME improves attribution robustness through its unique neighbor sampling approach for continuous features (Agarwal et al., 2021).

Notably, LIME can also be viewed through a perturbation lens, as it fundamentally perturbs input features to approximate the model's output. This connection points to a unification: many feature attribution methods can be understood within a common mathematical framework of local function approximation, which we explore next.

### 3.4 Unifying Feature Attributions via Local Function Approximation

While the original algorithms of feature attribution methods discussed above can be viewed in their respective three categories, many of them can be unified under a common local function approximation framework (Han et al., 2022). Within this framework, a model $f$ is approximated around a point of interest $x$ in a local neighborhood distribution $\mathcal{Z}$ by an interpretable model $g$ using a loss function $\ell$. Han et al. (2022) show that eight prominent feature attribution methods (Occlusion, KernelSHAP, Vanilla Gradients, Gradients $\times$ Input, Integrated Gradients, SmoothGrad, LIME, and C-LIME) can be viewed as specific instances of this framework, distinguished only by their unique choices of local neighborhoods $\mathcal{Z}$ and loss functions $\ell$ (Appendix Table 3).

The local function approximation framework (Han et al., 2022) enhances our understanding of feature attribution methods in several important ways. First, it provides conceptual coherence to the field. While different methods appear to have distinct motivations, this framework reveals their shared fundamental goal of local function approximation. Second, placing diverse methods under a single framework enables direct comparisons among them. This comparative lens allows us to better understand their similarities, differences, and behavior, such as why different methods sometimes generate disagreeing or even contradictory explanations for the same model prediction (Krishna* et al., 2024). Third, this unification enables theoretical simplicity. Instead of studying methods seperately, theoretical analyses can be performed using the framework and applied to each method, as shown by the no free lunch theorem and guiding principle in Han et al. (2022). Fourth, the conceptual understanding brought about by unification leads to principled, practical recommendations (Han et al., 2022). Additional details on this unification are provided in Appendix C.6.

## 4 Understanding Data Attributions

Data attribution studies how the training dataset $\mathcal{D}_{\text{train}}$ shapes model behavior. These methods are also known as *data valuation*, as they help assess the value of data from vendors and content creators. For each training example $x^{(j)}$, an attribution score $\psi_j(x)$ traces back to the training phase to quantify its influence on the model's output $f(x)$ for a test point $x$. These scores characterize training data properties, help identify mislabeled data, and justify training data values. Like feature attribution, data attribution methods can be organized into three categories: perturbation-based methods, gradient-based methods, and linear approximation methods. We examine prominent methods from each category below, with additional details in Appendix D.

### 4.1 Perturbation-Based Data Attribution

Perturbation-based data attribution observes the model behavior changes after removing or re-weighting the training data points and subsequently retraining the model, so these methods are also referred to as *retraining-based methods* (Hammoudeh & Lowd, 2024).

**Leave-One-Out (LOO) Attribution** is a prominent example of this approach, analogous to direct perturbation in feature attribution. The method trains a model on the complete dataset and then

separately removes each individual data point and retrains the model. The attribution score for each removed point is determined by the difference in performance between the original and retrained models. The LOO approach has a long history in statistics (Cook & Weisberg, 1982) and has proven valuable for modern AI model data attribution (Jia et al., 2021). It provides valuable counterfactual insights with its main limitation being computational cost, as it requires retraining the model for each data point. Many newer attribution methods can be viewed as efficient approximations of LOO. A natural extension of LOO is to leave a set of data points out to evaluate their collective impact through retraining (Ilyas et al., 2022).

**Game-Theoretic Data Attribution** represents the successful application of game theory to quantify training sample influence similar to feature attribution. As a direct perturbation method, LOO attribution overlooks interactions between data points, potentially missing subtle influence behaviors (Lin et al., 2022; Jia et al., 2021). Game-theoretic data attribution methods address this by treating training data points as players in a cooperative game, aiming to fairly distribute the model's performance among training samples. Data Shapley (Ghorbani & Zou, 2019) first applied Shapley values to data attribution by computing each training point's aggregated marginal contribution across all possible training data subsets. Although theoretically sound, game-theoretic methods face prohibitive computational costs for large datasets, as each marginal contribution requires model retraining and there are $2^n$ possible subsets. Various approximation methods have been proposed to address this challenge, which we discuss in Appendix D.2.

## 4.2 GRADIENT-BASED DATA ATTRIBUTION

Gradient-based data attribution methods leverage the gradients of the loss with respect to training data $\nabla_\theta \mathcal{L}(f_\theta(x^{(j)}))$ and test data $\nabla_\theta \mathcal{L}(f_\theta(x))$ to assess the impact of training points $x^{(j)}$ on model output $f(x)$. As in Charpiat et al. (2019), simple dot product (GradDot) and cosine similarity (Grad-Cos) between these two gradients are used as similarity measures and consequently attribution scores $\psi_j(x)$. Like feature attribution, gradient-based methods often offer greater computational efficiency than perturbation-based methods since they typically require no retraining.

**Influence Function (IF)**, a classic statistical technique originally developed for analyzing influential points in linear regression (Cook & Weisberg, 1980), has been adapted for modern AI models (Koh & Liang, 2017). IF approximates LOO model parameter changes by Taylor expansion, avoiding explicit retraining. This approximation builds on computing both the gradient and the (inverse) Hessian of the loss with respect to model parameters. IF offers an effective and computationally feasible alternative to LOO, but it also faces several challenges. Its convexity assumptions often do not hold for modern AI models, and its Hessian computation remains expensive for large models. Many methods have been proposed to address these limitations; we discuss IF and these enhancements in detail in Appendix D.3.

**Tracing (Training) Path** While many gradient-based methods follow IF to compute gradients at the final model parameters, TracIn (Pruthi et al., 2020) introduces a novel approach that traces the influence of training instances throughout the entire training process. The method attributes influence by computing dot products between training and test data gradients at each training step from the initial model parameters to the final model parameters at the end of training, accumulating these to capture a training point's total influence across the training path. This path tracing approach provides valuable insights into training dynamics while avoiding limitations of LOO and IF, such as assigning identical attribution scores to duplicate training data points. TracIn also offers greater flexibility than IF by eliminating the convexity assumption and Hessian matrix computations. On the other hand, its tracing requires storing intermediate model checkpoints during training, increasing both memory usage and computational costs.

## 4.3 LINEAR APPROXIMATION FOR DATA ATTRIBUTION

Datamodel (Ilyas et al., 2022) applies linear approximation to data attribution, similar to LIME in feature attribution. It constructs a linear model $g$ with $n$ coefficients and $\{0,1\}^n$ vectors as inputs, where each input represents a subset of training data. $g$ is learned to map any counterfactual subset of training data to output $f(x)$, where $f$ is trained on this subset with the given model architecture and training algorithm. The coefficients of $g$ thus represent the attribution scores of the training data points. The method's counterfactual nature enables evaluation of other attribution methods via the

*Linear Datamodeling Score (LDS)*, which compares their attribution score rankings to Datamodel's ranking. While Datamodel can effectively capture model behavior, constructing this large linear model requires extensive counterfactual data obtained by training model $f$ on various subsets, making it computationally intensive. TRAK (Park et al., 2023) addresses these computational challenges by estimating Datamodels in a transformed space where the learning problem becomes convex and can be approximated efficiently. It further improves efficiency through random projection of model parameters and ensemble attribution results of multiple trained models. Though the ensemble approach still requires some model retraining on different subsets, it achieves high estimation accuracy with significantly fewer retraining iterations than Datamodel. Furthermore, both approaches can be viewed as perturbation-based methods, similar to LIME, as they systematically vary training data to construct linear models.

## 5 UNDERSTANDING COMPONENT ATTRIBUTIONS

Component attribution, an emerging approach within *mechanistic interpretability*, seeks to understand AI models by reverse engineering their internal mechanisms into interpretable algorithms. Operating primarily at test time for model inference, it quantifies how each model component $c_k$ contributes to a model output $f(x)$ through an attribution score $\gamma_k(x)$. Components can be defined flexibly across different scales - from individual neurons and attention heads to entire layers and *circuits* (subnetworks). By identifying components responsible for specific behaviors, this approach enables deeper model understanding and targeted model editing. Like feature and data attribution, component attribution methods fall into three categories: perturbation-based, gradient-based, and linear approximation approaches. Below we examine key methods from each category, with additional details provided in Appendix E.

### 5.1 PERTURBATION-BASED METHODS

In component attribution, perturbation-based methods are fundamentally quite similar to perturbation-based methods in feature and data attribution. Components of the model, whether neurons, circuits, or layers, are similarly perturbed to measure their effect on model behavior. Generally, the perturbations are chosen carefully to attempt to localize behaviors related to specific tasks or concepts.

**Causal Mediation Analysis** (Pearl, 2022; Vig et al., 2020) is based on the abstraction of models to causal graphs. These graphs consist of nodes, which can be components such as neurons, circuits, attention heads, or layers, and directed edges that represent the causal relationships between nodes. Causal mediation analysis is defined by an input cause $x$ and an output effect $f(x)$ that is mediated by intermediate causal nodes between $x$ and $f(x)$. By perturbing these intermediate components, $c_k$, changes in $f(x)$ can be measured to get attribution scores $\gamma_k(x)$. These indirect effects are often measured counterfactually in order to calculate each component's contribution towards a particular behavior, such as a correct factual prediction. To do so, the activations of all intermediate components $c_k$ are measured during three separate runs: a clean run with no perturbations, a corrupted run where intermediate activations are perturbed, and a corrupted-with-restoration run that measures whether a single component can restore the prediction. The corrupted run can be repeated multiple times with different random noise added to obtain a more robust attribution score. This analysis is frequently referred to as *causal tracing* (Meng et al., 2022) or *activation patching*, and also *path patching* (Wang et al., 2022) when patching is applied to paths connecting components. By comparing the outputs of the clean and corrupted run, or by looking at the corrupted-with-restoration run, one is able to find the specific mediator components that are either sufficient or necessary for the desired behavior. By changing the dataset, metric, and causal mediators, we can model the relationship between each component and various model behaviors.

**Game-Theoretic Component Attribution** (Ghorbani & Zou, 2020) follows a similar approach to game-theoretic methods in feature and data attribution to quantify the contributions of each neuron to the model's performance. These methods take into account the interactions between neurons by modeling neurons as players in a cooperative game to fairly distribute contributions. In particular, Neuron Shapley (Ghorbani & Zou, 2020) extends prior works on Shapley values to component attribution, ensuring computational feasibility through sampling-based approximations and a multi-armed bandit algorithm that efficiently identifies neurons with large attribution scores.

**Mask Learning and Subnetwork Probing** (Csordás et al., 2020; Cao et al., 2021) adopts a similar concept to feature attribution, attempting to approximate either the model's or a probe's performance on a given task by searching for a subnetwork or components that equivalently perform that task. More specifically, subnetwork probing optimizes a mask for the weights of the model, essentially pruning the model, by performing gradient descent on a continuous relaxation of searching for the subnetwork of a model that performs the task of interest. Thus, behavior can be attributed to the parts of the network that are not masked out.

## 5.2 GRADIENT-BASED COMPONENT ATTRIBUTION

To further decrease the computational complexity of component attribution methods, researchers have developed alterations of causal tracing that leverage gradient-based approximations requiring only two forward passes and a single backward pass to generate attributions. Attribution patching (Nanda, 2023) is the simplest gradient-based approximation of causal tracing. Intuitively, attribution patching leverages a linear approximation of the model for the corrupted prompts and measures the local change when patching a single activation from the corrupted to clean input. This is achieved by computing the backward pass for the corrupted output with respect to the patching metric and storing the gradients with respect to the activations. Note that for feature and data attributions, gradients are taken with respect to the input features or training data, not the model activations. Finally, the method takes the difference between the clean and corrupted activations and multiplies it by the cached gradients to obtain attribution scores.

## 5.3 LINEAR APPROXIMATION FOR COMPONENT ATTRIBUTION

Given the rapid increase in model size and the combinatorial nature of searching for effective components, component attribution also employs linear approximations like LIME and Datamodels. COAR (Shah et al., 2024) attempts to decompose model behavior in terms of various model components by predicting the counterfactual impact of ablating each component, similar to many forms of causal mediation analysis. Given the computational complexity of this problem, they employ linear approximations by assigning scores to each component of a model and estimating the counterfactual effect of removing sets of components by simply summing their corresponding scores. Thus, the complexity of relationships between components is abstracted away through the linear approximation.

## 6 POSITION AND CONTRIBUTIONS

Feature, data, and component attribution methods have largely been studied as separate problems, resulting in the parallel development of similar methods from different communities with distinct terminologies. We argue that these methods can be unified into a holistic view. Having demonstrated their methodological similarities across three types of attribution, we now summarize their common concepts and challenges and identify promising research directions through cross-aspect knowledge transfer. We believe that our unified view will bridge the current fragmented landscape, make the field more accessible to newcomers, and help advance research in interpretability and beyond.

### 6.1 COMMON CONCEPTS OF ATTRIBUTION METHODS

As we discussed in the previous sections, attribution methods across features, data, and components can be categorized into: perturbation-based, gradient-based, and linear approximation. We provide a detailed discussion of these categories in Appendix B with Table 2 summarizing all the methods we discussed. Beyond algorithmic similarities, conceptual ideas also transfer across aspects of attribution. One example is the deliberate introduction of randomness and smoothing to enhance attribution robustness. This idea has proven effective across three attribution types through random noisy samples used by SmoothGrad for feature attribution, attribution results ensembled over multiple retrainings by TRAK for data attribution, and aggregated results from multiple corrupted runs in causal mediation analysis for component attribution. Another example is tracking and aggregating results along paths, as Integrated Gradients along paths from base input to target input in feature attribution, TracIn tracing training paths to reveal dynamic data influences in data attribution, and path

patching tracking component effects along residual stream paths in component attribution. These shared concepts highlight the fundamental connections of attribution methods.

## 6.2 COMMON CHALLENGES OF ATTRIBUTION METHODS

Attribution methods also face common challenges that impact their reliability and practical utility, which we briefly discuss below and extend in Appendix F.

**Computational Challenges** present substantial barriers preventing attribution methods from being applied to large models. These challenges appear in all three types of attributions, rooted in their shared technical methods. For perturbation-based methods, the curse of dimensionality makes it intractable to comprehensively analyze high-dimensional inputs, large training datasets, and models with numerous components. Gradient-based methods offer more practical computational costs, except when sophisticated gradient computations are needed, such as aggregating mang gradients or computing second-order Hessian matrices. Linear approximation methods also face challenges when numerous data points and model evaluations are required to establish sufficient data for learning accurate linear models.

**Consistency Challenges** refer to the variability in attribution results across multiple runs of the same method with different random seeds, making it challenging to establish stable interpretations and evaluations. This challenge is prevalent across all three types of attribution due to multiple sources of randomness in their common techniques, including sampling, learning processes with stochastic optimization, and also non-trivial hyperparameters. While some gradient-based methods can produce consistent results in a single run when they do not involve sampling or approximations for computationally intensive operations, the consistency between different gradient-based methods varies considerably, which is a problem for all three method categories and leads to the following evaluation challenges.

**Evaluation Challenges** arise in all three types of attribution due to multiple factors. Inconsistent results make it difficult to reliably compare methods and determine their relative accuracy. This challenge is particularly evident in gradient-based feature attribution methods. As demonstrated by Adebayo et al. (2018), they can produce contradictory attribution results and sometimes perform no better than random baselines, making them difficult to evaluate fairly. For existing evaluation metrics, counterfactual evaluation could provide more rigorous validation, but computational constraints often make this approach impractical. Task-specific evaluations offer easier alternatives but frequently lack generalizability across different contexts. Human evaluation, despite being considered the gold standard, faces scalability issues and potential biases. The diversity of evaluation metrics and their varying definitions of importance make the evaluation more challenging. An attribution result may perform well by one metric but poorly according to another. These challenges emphasize the pressing need for developing more reliable and practical attribution evaluation metrics.

## 6.3 CROSS-ASPECT ATTRIBUTION INNOVATION

The connections among feature, data, and component attributions discussed in the sections above suggest multiple promising directions for future research. One research direction is to leverage insights from one type of attribution to develop methods for another. This can be directly identified as filling in the empty cells in Table 2. For example, while the Shapley value has been successfully applied across all three types of attribution, many other game-theoretic notions have only been used in feature attribution and not for data and component attribution. In addition, some advanced gradient techniques are common for feature and data attribution methods, but not for component attribution. The Hessian matrix, for example, has been used to obtain second-order information in Integrated Hessians for feature attribution and extensively in all IF-related methods for data attribution, and can be explored for component attribution.

Moreover, seeing the theoretical connections among feature, data, and component attributions enables us to draw inspiration from one area to advance our understanding of another as a whole. For example, we demonstrated that diverse feature attribution methods all perform local function approximation (§ 3.4). This framework can potentially also apply to data and component attributions. We know that feature attributions perform function approximation of the blackbox model's predic-

tions over the space of input features. One may hypothesize that data attributions perform function approximation of the model's weights over the space of training data points and that component attributions perform function approximation of the model's predictions over the space of model components. If so, function approximation may unify data and component attribution methods as well. Such theoretical unification may provide plentiful benefits to data and component attribution, including conceptual coherence, elucidation of method properties, theoretical simplicity, and clearer practical recommendations.

Another research direction is to move towards more holistic analyses of model behavior. These attribution methods provide insight into model behavior through different lenses: input features, training data, and model components. Each type of attribution provides different and complementary information about model behavior. For example, for a given model prediction, feature attributions may not suggest that the model is relying on sensitive features to make predictions, but model component attributions may uncover a set of neurons that encode biased patterns. In this sense, focusing only on one type of attribution, i.e., studying only one part of the model, is insufficient to understand model behavior. Thus, future research may develop approaches to enable more comprehensive model understanding, such as understanding how to use different types of attribution methods together, the settings under which different attribution types may support or contradict one another, and the interactions between the three model parts (e.g., how patterns in the training data are encoded in model neurons).

## 6.4 CONNECTIONS TO OTHER AREAS OF AI

Attribution methods also hold immense potential to benefit other AI areas. Especially with a unified view integrating feature, data, and component attribution, researchers can not only gain deeper insights of model behavior but also edit and steer models towards desired goals and improve model compliance with regulatory standards.

**Model Editing** (De Cao et al., 2021; Mitchell et al., 2021; Meng et al., 2022, inter alia) focuses on precisely modifying models without retraining. It enables researchers to correct model mistakes, analogous to fixing bugs in software. This approach is particularly valuable for large language models (LLMs), which encode vast information in their parameters and are prohibitively expensive to retrain. It can be viewed as a downstream task of attribution methods. Once attribution methods locate an issue, editing methods can be applied to the problematic parts. While editing aligns most closely with component attribution, other attribution types serve essential complementary functions. Feature attribution identifies spurious correlations requiring correction, and data attribution reveals problematic training samples that influence model behavior. The unified attribution framework provides a holistic perspective that enables more efficient and accurate editing, especially when component attribution alone proves insufficient (Hase et al., 2024).

**Model Steering** (Zou et al., 2023, inter alia) differs from model editing by integrating a steering vector into the model's inference process rather than modifying model parameters. While editing focuses on specific knowledge modifications, steering guides model behavior at a higher level, such as enhancing truthfulness and harmlessness in LLMs. Similar to model editing, a unified attribution framework can significantly enhance steering by better localizing target components to steer and generating more effective steering vectors through relevant features and training data.

**Model Regulation** (Oesterling et al., 2024, inter alia) is an emerging field examining the relationship between AI systems, policy, and societal outcomes. Regulation and policy frequently stress the need for transparency of AI systems as well as users' right to an explanation. Attribution methods provide an avenue for practitioners to ensure that AI systems meet these legal and ethical requirements, by providing information about the overall AI system as well as specific input-output behavior. Feature attribution reveals input processing patterns, data attribution exposes training data influences, and component attribution illuminates architectural roles. This multi-faceted understanding enables more targeted and effective regulation. For example, when addressing biased behavior, feature attribution can be used to identify discriminatory input patterns, data attribution to trace problematic training samples or copyright infringements, and component attribution to locate architectural elements needing adjustment. These complementary perspectives provide the comprehensive understanding needed to guide model regulation toward desired societal outcomes.

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

APPENDIX

## A  SUMMARY OF NOTATIONS

In Table 1, we summarize the notation used in this paper.

Table 1: Summary of notations.

| Notation | Description |
|---|---|
| $\mathcal{D}_{\text{train}}$ | Training dataset $\{x^{(1)}, \cdots, x^{(n)}\}$ |
| $f_\theta$/ $f$ | Model trained on $\mathcal{D}_{\text{train}}$, parameters $\theta$ may be omitted |
| $c$ | Internal model components $\{c_1, \cdots, c_m\}$, definition is method-specific |
| $x^{\text{test}}$/$x$ | Model input at test time for inference, superscript "test" may be omitted |
| $\phi_i(x)$ | Attribution score of input feature $x_i$ for model output $f(x)$ |
| $\psi_j(x)$ | Attribution score of training data point $x^{(j)}$ for model output $f(x)$ |
| $\gamma_k(x)$ | Attribution score of internal model component $c_k$ for model output $f(x)$ |
| $g$ | Attribution function, which provides attribution scores for elements |
| $\mathcal{L}$ | Loss function for training the model $f$ |
| $\ell$ | Loss function for learning the attribution function $g$ |

## B  SUMMARY OF METHODS

As we discussed in the main body, attribution methods across features, data, and components can be categorized into three main approaches: perturbation-based methods, gradient-based methods, and linear approximation methods. Perturbation-based methods measure how a model's output changes when modifying specific elements, whether they are input features, training data points, or model components. To capture interactions between multiple elements, all three types of attribution methods employ common mathematical tools, such as the Shapley value from game theory. Gradient-based methods analyze model behavior by leveraging gradients to provide insights into the model's sensitivity to small input changes. Gradients bridge model behavior and the elements we wish to attribute without perturbations. Attributions are achieved in different types of gradients: computing gradients of model outputs with respect to input features to quantify feature importance, calculating gradients of loss functions with respect to specific training data points to analyze data influence, or using gradients to approximate the effects of modifying model components. Linear approximation methods fit linear models to approximate complex model behaviors. The inputs to these linear models can be input features, training data points, or model components. In some cases, binary indicators replace the actual elements as inputs to simplify the approximation.

In total, thousands of attribution methods of all three types have been proposed making a comprehensive literature summary infeasible. In Table 2, we summarize the attribution methods discussed in this paper, which we believe are the representative ones and align with the unified view we presented. Most of the empty cells in Table 2 (labeled as "–") represent methods that we believe are promising but have not yet been explored in the literature, as discussed in § 6.3. These represent research ideas that have been verified in one attribution type but remain unexplored in others. The exception is mask learning for data attribution, which we consider less promising because learning a high-dimensional mask of size $n$ jointly with the model would be infeasible when the model has not been trained.

## C  DETAILED DISCUSSION OF FEATURE ATTRIBUTION METHODS

Feature attribution methods can be broadly classified into three categories. Perturbation-based methods attribute feature importance by observing changes in model output when input features are altered or removed. These methods provide intuitive results but can be computationally expensive for high-dimensional data. Gradient-based methods utilize the model's gradients with respect to input features to attribute their importance. These methods are popular for differentiable models like neural networks, as they are often computationally efficient. Linear approximation methods construct

Table 2: A summary of representative feature, data, and component attribution methods classified into three methodological categories demonstrating our unified view.

| Method | | Feature Attribution | Data Attribution | Component Attribution |
|---|---|---|---|---|
| **Perturb** | Direct | Occlusions (Zeiler & Fergus, 2014)
RISE (Petsiuk, 2018) | LOO (Cook & Weisberg, 1982) | Causal Tracing (Meng et al., 2022)
Path Patching (Wang et al., 2022)
Vig et al. (2020)
Bau et al. (2020)
ACDC (Conmy et al., 2023) |
| | Game-Theoretic (Shapley) | SHAP (Lundberg & Lee, 2017) | Data Shapley (Ghorbani & Zou, 2019)
TMC Shapley (Ghorbani & Zou, 2019)
KNN Shapley (Jia et al., 2019)
Beta Shapley (Kwon & Zou, 2022) | Neuron Shapley (Ghorbani & Zou, 2020) |
| | Game-Theoretic (Others) | STII (Dhamdhere et al., 2019)
BII (Patel et al., 2021)
Core Value (Yan & Procaccia, 2021)
Myerson Value (Chen et al., 2018b)
HN Value (Zhang et al., 2022) | Data Banzhaf (Wang & Jia, 2023) | – |
| | Mask Learning | Dabkowski & Gal (2017)
L2X (Chen et al., 2018a) | – | Csordás et al. (2020)
Subnetwork Pruning (Cao et al., 2021) |
| **Gradient** | First-Order | Vanilla Gradients (Simonyan et al., 2013)
Gradient × Input (Shrikumar et al., 2017)
SmoothGrad (Smilkov et al., 2017)
GBP (Springenberg et al., 2014)
Grad-CAM (Selvaraju et al., 2016) | GradDot/GradCos (Pruthi et al., 2020) | Attribution Patching (Nanda, 2023)
EAP (Syed et al., 2023) |
| | Second-Order (Hessian/IF) | Integrated Hessian (Janizek et al., 2021) | IF (Koh & Liang, 2017)
FastIF (Guo et al., 2021)
Arnoldi IF (Schioppa et al., 2022)
EK-FAC (Grosse et al., 2023)
RelateIF (Barshan et al., 2020) | – |
| | Tracing Path | Integrated Grad (Sundararajan et al., 2017) | TracIn (Pruthi et al., 2020)
SGD-Influence (Hara et al., 2019)
SOURCE (Bae et al., 2024) | Attribution Path Patching (Nanda, 2023) |
| **Linear** | | LIME (Ribeiro et al., 2016)
C-LIME (Agarwal et al., 2021) | Datamodels (Ilyas et al., 2022)
TRAK (Park et al., 2023) | COAR (Shah et al., 2024) |

interpretable linear models of input features that approximate the behavior of the original complex model in the vicinity of a specific input and compute attribution scores from the linear model coefficients. These methods offer a balance between interpretability and local accuracy. Each category of methods has its strengths and limitations, making the choice of method dependent on the specific model, data characteristics, and attribution requirements of the task at hand. We now extend the discussion of some methods mentioned in the main text in more detail and provide discussions of some additional methods.

## C.1 DIRECT PERTURBATION FOR FEATURE ATTRIBUTION

**RISE** (Petsiuk, 2018) is a direct perturbation method that addresses limitations of earlier methods like occlusion while expanding applicability to complex models. The method provides a systematic approach for assessing feature importance through efficient sampling and aggregation of perturbations. It operates by randomly masking different regions of the input image and measuring the model's output to each masked version. The final saliency map is constructed by combining these random masks, with each mask weighted according to the model's predicted probability on the corresponding masked input. This sampling-based approach allows RISE to efficiently estimate feature importance while capturing interactions between different image regions.

## C.2 GAME-THEORETIC FEATURE ATTRIBUTION

The Shapley value, a solution concept from cooperative game theory introduced by Lloyd Shapley (Shapley, 1953), has gained particular prominence in feature attribution. For a data point $x$ with features $\{x_1, x_2, \ldots, x_d\}$, the Shapley value of feature $x_i$ for the model prediction $f(x)$ is defined as:

$$\phi_i(x, f) = \sum_{x_S \subseteq x \setminus \{x_i\}} \frac{1}{\binom{d-1}{|S|}} [f(x_S \cup \{x_i\}) - f(x_S)]$$

where $x_S$ represents a subset of features excluding feature $x_i$ indexed by $S$, and $f(x_S)$ denotes the model's prediction when only features in set $x_S$ are present. $f(x_S \cup \{x_i\}) - f(x_S)$ is the marginal

contribution of feature $x_i$ to the subset $x_S$ for the model's prediction. The formula computes the average marginal contribution of feature $x_i$ across all possible feature subsets. We simplify the attribution score notation by writing $\phi_i(x) = \phi_i(x, f)$.

Shapley values possess several desirable properties that make them particularly suitable for feature attribution:

- Efficiency: The attributions sum to the total prediction, i.e., $\sum_i \phi_i(x, f) = f(x) - f(\emptyset)$
- Symmetry: Features that contribute equally receive equal attribution, i.e., if $f(x_S \cup \{x_i\}) - f(x_S) = f(x_T \cup \{x_j\}) - f(x_T)$ for all subsets $x_S, x_T \subseteq x$, then $\phi_i(x, f) = \phi_j(x, f)$
- Linearity: For models $f_1$ and $f_2$, $\phi_i(x, a_1 f_1 + a_2 f_2) = a_1 \phi_i(x, f_1) + a_2 \phi_i(x, f_2)$ for constants $a_1$ and $a_2$.
- Null player: Features that don't affect the prediction receive zero attribution, i.e., if $f(x_S \cup \{x_i\}) - f(x_S) = 0$ for all subsets $x_S \subseteq x$, then $\phi_i(x, f) = 0$

These properties offer theoretical guarantees for fair and consistent feature attribution, making Shapley values a principled approach to understanding model behavior. However, the exact computation requires evaluating $2^{|x|}$ feature combinations, leading to various approximation methods in practice.

**Other Game-Theoretic Concepts**   Besides the Shapley value, other cooperative game-theoretic concepts are also applicable to feature attribution, offering different trade-offs between computational complexity and specific properties of the resulting attributions. The Shapley Taylor Interaction Index (STII) (Dhamdhere et al., 2019) is another concept that can be used for feature attribution, which is a generalization of the Shapley value that explicitly considers interactions between features. The Banzhaf Interaction Index (BII) (Patel et al., 2021) is particularly useful for considering joint feature interactions with simpler computation than the Shapley value. The core value (Yan & Procaccia, 2021), for instance, employs different axioms and emphasizes attribution stability. Additionally, the Myerson value (Chen et al., 2018b) and HN Value (Zhang et al., 2022) are valuable when prior knowledge about the feature structure is available. These alternative approaches provide researchers and practitioners with a range of tools to tailor their feature attribution methods to specific needs and constraints of their models and datasets.

**Connection to Linear Approximation**   The most common Shapley value-based attribution method, SHAP (Lundberg & Lee, 2017), is a perturbation-based method rooted in cooperative game theory. However, it can also be viewed through the lens of linear approximation methods, representing a unified approach of local linear attribution and classic Shapley value estimation. In the context of linear approximation, SHAP can be interpreted as fitting a linear model where features are players in a cooperative game, and the model output is the game's payoff. The SHAP framework includes variants like Kernel SHAP, which uses a specially kernel for weighted local linear regression to estimate SHAP values, effectively approximating the model's behavior in the feature space surrounding the instance being explained. This perspective on SHAP highlights its connection to linear approximation methods while retaining its game-theoretic foundations.

## C.3   Perturbation Mask Learning for Feature Attribution

Mask learning methods offer several notable advantages in feature attribution. They provide attributions more efficiently, especially for high-dimensional inputs, which is particularly beneficial for complex models and large datasets. The continuous spectrum of importance scores generated by these techniques offers more nuanced insights than binary approaches, allowing for a finer-grained understanding of feature relevance. Furthermore, the learning process can implicitly capture complex feature interactions, providing a more comprehensive view of how features contribute to model decisions. Additionally, these methods can be tailored to specific model architectures and incorporate domain-specific constraints, enhancing their flexibility and applicability across various fields.

**L2X**   (Chen et al., 2018a) frames feature attribution as an optimization problem and learns a masking model to generate masks that maximize mutual information between input feature subsets and model output. This approach not only identifies important features but also captures their interdependencies, providing a comprehensive explanation of the model behavior. L2X is versatile and

applicable to various domains beyond image classification. For instance, it has been successfully applied to sentiment classification tasks using datasets of movie reviews.

**Gradient Computation in Mask Learning vs. Gradient-Based Methods**  To avoid potential confusion, it is important to note that while mask learning methods may utilize gradient computation during the learning process, these gradients serve a different purpose than those in gradient-based attribution methods. In mask learning, gradients are used to learn a soft mask or an explainer model for generating masks. These gradients are not used to directly determine the feature attribution scores themselves. This distinction sets mask learning approaches apart from the gradient-based methods discussed in the following section.

## C.4 GRADIENT-BASED FEATURE ATTRIBUTION

Gradients for feature attribution are very different from those used in model training. For a model $f$ with parameters $\theta$, gradients of the loss function are taken with respect to the parameters ($\nabla_\theta \mathcal{L}(\theta)$) to guide parameter updates during training. In feature attribution, gradients of the model's output $f(x)$ are taken with respect to input features ($\nabla_x f(x)$) to quantify each feature's contribution to the model's output. Gradient-based methods have been widely adopted for feature attribution because of their computational efficiency. They typically require only a single forward and backward pass through the model to compute the gradients, and require no additional perturbation or linear model fitting, which makes them particularly suitable for real-time applications and large-scale datasets. On the other hand, gradient-based methods have two key limitations. They require access to model parameters and only work with differentiable models. Additionally, the gradient results can be nonrobust as we discussed in Section 6.2. Despite these challenges, gradient-based methods remain a fundamental tool in the feature attribution toolkit.

**Gradient × Input**  (Shrikumar et al., 2017) improves over vanilla gradients. By multiplying the input features element-wise with their corresponding gradients, this method mitigates the "gradient saturation" problem where gradients can become very small even for important features. The element-wise multiplication also helps reduce visual diffusion in the attribution score visualization, resulting in sharper and more focused visualizations of important features.

**Integrated Gradients**  (Sundararajan et al., 2017) provides a theoretically grounded approach to feature attribution by accumulating gradients along a path from a baseline input to the actual input. This method satisfies important axioms including sensitivity (a change in input leads to a change in attribution) and implementation invariance (attributions are identical for functionally equivalent networks). The integration process captures the cumulative effect of each feature as it transitions from the baseline to its actual value, providing a more complete picture of feature importance than vanilla gradients of a single input.

**Integrated Hessians**  (Janizek et al., 2021) extends the integrated gradients method to analyze feature interactions, with the goal of understanding how features interact. This method treats the integrated gradient function as differentiable and quantifies interactions between two features using second-order information, the Hessian matrix. By computing these Hessian-based interactions along the same integration path used in integrated gradients, it provides a principled way to measure feature interdependencies and more comprehensive feature attributions.

**Guided Backpropagation (GBP)**  (Springenberg et al., 2014) modifies the standard backpropagation process of NNs to generate cleaner and more interpretable attribution results. When propagating gradients through ReLU units, GBP sets negative gradient entries to zero, effectively combining the signal from both the higher layer and the ReLU units. This modification helps eliminate artifacts and noise in the attribution results while preserving the positive contributions of features, resulting in sharper and more visually interpretable feature attributions.

**Grad-CAM**  (Selvaraju et al., 2016) is a widely used method for attribution and visualization of important regions in images, specifically for convolutional NNs. It computes the gradient of the target class score (logit) with respect to the feature maps in the last convolutional layer. These gradients are then used as weights to combine the feature maps, creating a coarse localization map

that highlights important regions for predicting the target class. The resulting localization map is upsampled to match the input image size to create a saliency map, providing an interpretable visualization of features important for the model's output.

**Generalizing Gradient-Based Feature Attribution** Several additional methods share similar underlying principles with gradient-based approaches, although they do not directly compute gradients in their original formulation. For example, Layer-wise Relevance Propagation (LRP) (Bach et al., 2015) propagates predictions backwards through the network while preserving the total relevance at each layer. LRP provides a unique perspective on attribution by focusing on the relevance of individual neurons to the final prediction. Similarly, DeepLIFT (Shrikumar et al., 2017) operates by comparing each neuron's activation to a reference activation and propagating the resulting differences to the input features. Interestingly, Ancona et al. (2017) demonstrated that for ReLU networks with zero baseline and no biases, both $\epsilon$-LRP and DeepLIFT (rescale) methods are mathematically equivalent to the Input $\times$ Gradient approach. For a more detailed analysis of these equivalences, we refer readers to Ancona et al. (2017).

### C.5 Linear Approximation for Feature Attribution

**C-LIME** (Agarwal et al., 2021) is a variant of LIME specifically designed for continuous features that generates local explanations by sampling inputs in the neighborhood of a given point. It differs from LIME in several aspects: it uses a constant distance metric and Gaussian sampling centered at the input point rather than uniform random sampling, making perturbations naturally closer to the input without requiring explicit weighting. C-LIME also restricts itself to linear models for continuous features, unlike LIME's more general model class, and excludes regularization by setting the regularizer term to zero. For simplicity, C-LIME focuses on feature weights while ignoring the intercept terms.

**Generalizing Linear Approximations for Feature Attribution** For both LIME, C-LIME, and other similar linear approximation methods, the assumption that the model's behavior can be reasonably approximated by a linear function in the local neighborhood is crucial. In this context, while the linear model serves as a proxy, it can be replaced by a more complex yet interpretable model that is still capable of providing attribution results. For instance, LIMETree uses tree models (Sokol & Flach, 2020). We refer readers to Sokol & Flach (2020) for a detailed discussion of this approach.

### C.6 Unifying Feature Attribution Methods Through Local Function Approximation

Under the local function approximation framework, the model $f$ is approximated by an interpretable model class $\mathcal{G}$ around the point of interest $x$ over a local neighborhood distribution $\mathcal{Z}$ using a loss function $\ell$. The approximation is given by

$$g^* = \arg\min_{g \in \mathcal{G}} \mathbb{E}_{\xi \sim \mathcal{Z}} \ell(f, g, x, \xi).$$

Han et al. (2022) show that at least eight feature attribution methods (Occlusion, KernelSHAP, Vanilla Gradients, Gradients $\times$ Input, Integrated Gradients, SmoothGrad, LIME, and C-LIME) are all instances of this framework. These methods all use the linear model class $\mathcal{G}$ to approximate $f$, but do so over different local neighborhoods $\mathcal{Z}$ using different loss functions $\ell$ as in Table 3.

Under this setup, $g$'s model weights are equivalent to the explanation obtained using each method's original algorithm. Also, note that for the local function approximation framework, there are requirements on the loss function: a valid loss $\ell$ is one such that $\mathbb{E}_{\xi \sim \mathcal{Z}} \ell(f, g, x, \xi) = 0 \iff f(x^{\{\xi\}}) = g(x^{\{\xi\}}) \ \forall \xi \sim \mathcal{Z}$.

In addition, while the local function approximation framework may seem similar to LIME, it differs from LIME by 1) requiring that $f$ and $g$ share in the same input and output domain, 2) imposing the condition on the loss function $\ell$ discussed above, and 3) following the standard machine learning methodology to avoid overfitting and to tune hyperparameters. A more detailed discussion can be found in Section 3 of Han et al. (2022).

Table 3: Existing methods perform local function approximation of a black-box model $f$ using the interpretable model class $\mathcal{G}$ of linear models where $g(x) = w^\top x$ over a local neighbourhood $\mathcal{Z}$ around point $x$ based on a loss function $\ell$. $\odot$ indicates element-wise multiplication. (Table reproduced from Han et al. (2022)).

| Techniques | Attribution Methods | Local Neighborhood $\mathcal{Z}$ around $x^{\{0\}}$ | Loss Function $\ell$ |
|---|---|---|---|
| Perturbations | Occlusion | $x \odot \xi$; $\xi(\in \{0,1\}^d) \sim$ Random one-hot vectors | Squared Error |
| | KernelSHAP | $x^{\{0\}} \odot \xi$; $\xi(\in \{0,1\}^d) \sim$ Shapley kernel | Squared Error |
| Gradients | Vanilla Gradients | $x + \xi$; $\xi(\in \mathbb{R}^d) \sim$ Normal$(0, \sigma^2), \sigma \to 0$ | Gradient Matching |
| | Integrated Gradients | $\xi x$; $\xi(\in \mathbb{R}) \sim$ Uniform$(0,1)$ | Gradient Matching |
| | Gradients $\times$ Input | $\xi x$; $\xi(\in \mathbb{R}) \sim$ Uniform$(a,1), a \to 1$ | Gradient Matching |
| | SmoothGrad | $x + \xi$; $\xi(\in \mathbb{R}^d) \sim$ Normal$(0, \sigma^2)$ | Gradient Matching |
| Linear Approximations | LIME | $x \odot \xi$; $\xi(\in \{0,1\}^d) \sim$ Exponential kernel | Squared Error |
| | C-LIME | $x + \xi$; $\xi(\in \mathbb{R}^d) \sim$ Normal$(0, \sigma^2)$ | Squared Error |

# D  DETAILED DISCUSSION OF DATA ATTRIBUTION METHODS

Unlike feature attribution, data attribution methods trace back to the training phase and quantifies the training data's influence on the model's output, but they similarly fall into three categories. Perturbation-based methods assess training data importance by observing changes in model behavior when training samples are removed or modified. These methods provide accurate results but can be computationally expensive as they require retraining the model multiple times. Complete retraining-based methods like LOO are often used as a ground truth for evaluating other data attribution methods. Gradient-based methods utilize the model's gradients evaluated at the training data points and the test data point to quantify the influence of the training data points on the test data point. These methods avoid the computational cost of retraining the model but may face the challenges like the non-convexity of the loss landscape or the difficulty in computing the Hessian matrix efficiently. Linear approximation methods construct interpretable models that approximate how training data affects model behavior. The linear model operates on the entire training dataset, which can be suprisingly accurate but also heavy to train. Methods from different categories have their own strengths, limitations, and use cases. We now extend the discussion of some prominent methods mentioned in the main text in more detail and provide discussions of some additional methods.

## D.1  LEAVE-ONE-OUT DATA ATTRIBUTION

Leave-One-Out is a prominent example of perturbation-based data attribution, but also a natural idea that existed for a long time in statistics. For example, it has been used as a resampling technique (e.g., jackknife resampling (Tukey, 1958)) to estimate the bias and variance of a statistic of interest (such as a regression coefficient). LOO has been used to detect influential data points for linear regression (Cook & Weisberg, 1982), for example, through Cook's distance (Cook, 1977). Until recently, LOO has been applied to modern AI models to attribute model performance to individual training data points (Jia et al., 2021). It provides valuable counterfactual insights with its main limitation being computational cost, as it requires retraining the model for each data point. Many newer attribution methods, like the gradient-based methods can be viewed as efficient approximations of LOO.

## D.2  GAME-THEORETIC DATA ATTRIBUTION

The primary limitation of game-theoretic methods is their prohibitive computational cost for large datasets, as they require numerous model retrainings over the powerset of the training data. To address this challenge and further improve method robustness, researchers have proposed various methods.

**Truncated Monte Carlo (TMC) Shapley**  (Ghorbani & Zou, 2019) approximates the Shapley value by adopting an equivalent definition of the Shapley value in terms of aggregating over data permutations instead of data subsets (Shapley, 1953). The method works by truncating the number of permutations sampled and the number of data points considered in each permutation, and align

that with model training. For each sampled permutation, it computes the marginal contribution of each data point by evaluating model performance with and without that point. The gradient information from these evaluations is then used to as an estimate of each point's marginal contribution. TMC Shapley significantly reduces computational cost while maintaining reasonable approximation accuracy of the exact Shapley values.

**KNN Shapley** (Jia et al., 2019) introduces an efficient approximation for Shapley values data attribution by using K-Nearest Neighbors (KNN) as a surrogate model instead of retraining the full model. For each test point, it first identifies its K nearest neighbors in the training set. Then, it computes Shapley values only considering these neighbors' contributions to the KNN prediction, rather than the original model's prediction. This localized computation dramatically reduces complexity from exponential to polynomial in the number of neighbors K. The method maintains good attribution quality since nearby training points typically have the biggest influence on a test point. The KNN approximation aligns better with the goal of estimating the value of data from the data vendor's perspective, and thus was named data valuation in the original paper.

**Beta Shapley** (Kwon & Zou, 2022) extends the standard Data Shapley framework by introducing a beta distribution to weight different subset sizes differently. The original Shapley value weights subsets according to their sizes. The $\beta$ parameter controls how much emphasis is placed on smaller versus larger subsets when computing marginal contributions. This generalization relaxes the efficiency axiom of classical Shapley values, which requires attributions to sum to the total model value. By allowing this flexibility, Beta Shapley can better handle noisy or corrupted training data by reducing their influence on the attribution scores. The method provides theoretical analysis showing how different $\beta$ values affect properties like noise robustness and estimation variance.

**Data Banzhaf** (Wang & Jia, 2023) adapts the Banzhaf value from cooperative game theory as an alternative to Shapley values for data attribution. The Banzhaf value considers the average marginal contribution of training point across all possible data subsets like the Shapley value, but weights these contributions differently. This weighting scheme leads to the largest possible safety margin, making the attribution more robust to data perturbations and noise. The method provides theoretical guarantees on this robustness and demonstrates empirically that it can better identify mislabeled or adversarial training examples compared to Data Shapley.

### D.3 INFLUENCE FUNCTION AND ITS VARIANTS

Influence functions provide a way to estimate how model parameters would change if we reweight or remove a training point, without having to retrain the model. Given a model with parameters $\theta$ trained by minimizing the empirical risk $\frac{1}{n} \sum_{i=1}^{n} \mathcal{L}(x^{(i)}, \theta)$ over the training, the IF approximates the change in parameters when upweighting a training point $x^{(j)}$ by $\epsilon$:

$$\theta_{\epsilon, x^{(j)}} = \arg\min_{\theta} \frac{1}{n} \sum_{i=1}^{n} \mathcal{L}(x^{(i)}, \theta) + \epsilon \mathcal{L}(x^{(j)}, \theta)$$

Under the assumption that the loss function $\mathcal{L}$ is twice-differentiable and strictly convex, a first-order Taylor expansion around the final optimal model parameters $\theta^*$ gives:

$$\mathcal{I}_{\text{up,params}}(x^{(j)}) = -H_{\theta^*}^{-1} \nabla_{\theta} \mathcal{L}(x^{(j)}, \theta^*)$$

where $H_{\theta^*} = \frac{1}{n} \sum_{i=1}^{n} \nabla_{\theta}^2 \mathcal{L}(x^{(i)}, \theta^*)$ is the Hessian and is by assumption positive definite. The influence of training point $x^{(j)}$ on the loss at test point $x^{\text{test}}$ is the effect of this infinitesimal $\epsilon$-upweighting on test point's risk:

$$\mathcal{I}_{\text{up,loss}}(x^{(j)}, x^{\text{test}}) = -\nabla_{\theta} \mathcal{L}(x^{\text{test}}, \theta^*)^{\top} H_{\theta^*}^{-1} \nabla_{\theta} \mathcal{L}(x^{(j)}, \theta^*)$$

The negative $\mathcal{I}_{\text{up,loss}}(x^{(j)}, x^{\text{test}})$ will be the data attribution score $\psi_j(x)$ on $x^{\text{test}}$ and it provides an efficient approximation to LOO retraining (Koh & Liang, 2017).

While effective in certain scenarios and computationally more feasible than retraining-based methods, IF faces several challenges. First, it assumes convexity and double-differentiability, which are often not satisfied in deep learning scenarios. Second, it involves Hessian matrix computation, which

can be computationally expensive for large models. Also, the potential non-positive definiteness of the Hessian matrix in certain cases can lead to inaccuracies, often necessitating the introduction of dampening factors that may affect the precision of influence estimates. To address these limitations, many methods have been proposed to enhance the efficiency and applicability of IF.

**FastIF** (Guo et al., 2021) introduces several key optimizations to make IF more computationally tractable. First, it uses KNN to reduce the search space from the entire training set to a smaller subset of promising candidates that are likely to be influential. Second, it develops a fast estimation technique for the inverse Hessian-vector product that avoids computing and storing the full Hessian matrix and its inverse. Third, it implements parallelization strategies to asynchronously compute Hessian-vector products across multiple processors. These optimizations together enable FastIF to scale to much larger datasets while maintaining attribution quality comparable to the original IF.

**Arnoldi IF** (Schioppa et al., 2022) employs Arnoldi's iterative algorithm to efficiently identify the dominant eigenvalues and eigenvectors of the Hessian matrix. These dominant components serve as the basis for projecting all gradient vectors into a lower-dimensional subspace. Compute IF in this subspace substantially reduces the computational complexity. The method can be flexible by selecting an appropriate number of eigenvalues to retain. Empirical results demonstrate that this approach can achieve comparable attribution quality to full IF while significantly reducing both memory requirements and computation time.

**EK-FAC** (Grosse et al., 2023) leverages the Eigenvalue-corrected Kronecker-Factored Approximate Curvature (EK-FAC) parameterization to efficiently approximate the Hessian matrix. This parameterization exploits the natural block structure present in NNs to decompose the Hessian into more manageable components, which significantly reduces the computational complexity of Hessian-vector products. By leveraging these techniques, IF can be effectively scaled to large transformer models with hundreds of millions of parameters, which are orders of magnitude more complex than the simpler NNs originally considered by Koh & Liang (2017). Theoretical guarantees for the approximation quality and demonstrations of empirical success on foundation models were shown.

**RelateIF** (Barshan et al., 2020) addresses another limitation of IF other than their computational cost. Standard IF methods often highlight outliers or mislabeled data points as most influential, which may not always align with intuitive notions of influence. RelateIF introduces a novel approach that distinguishes between global and local influence by examining how training data affect specific predictions relative to their overall impact on the model. This relative influence measure helps identify training data points that have significant local influence on particular test predictions while accounting for their broader effects on the model. RelateIF better captures intuitive notions of influence while being more robust to outliers in the training data.

### D.4 TRACING TRAINING PATH FOR DATA ATTRIBUTION

Tracing training path for data attribution provides valuable insights of training dynamics while avoiding limitations of LOO and IF. Besides TracIn, there are other methods that trace training dynamics that provide more accurate attribution results but they are all more computationally expensive than those considering only final model parameters like IF.

**SGD-Influence** (Hara et al., 2019) traces the training path by approximating the training process with a series of unrolled steps to estimate data influence. The method estimates LOO influence by unrolling gradient descent using empirical risk Hessians, under the assumption that both the model and loss function are convex and the optimization algorithm is Stochastic Gradient Descent (SGD). SGD-Influence primarily applies unrolling to quantify the Cook's distance (Cook, 1977) between model parameters with and without a specific training point. To better align with attribution estimation, a surrogate linear influence estimator is used to incrementally update throughout the unrolling process. However, this approach requires unrolling the full training path for each test instance individually, which has significant computational complexity.

**SOURCE**  (Bae et al., 2024) extends training path tracing to better capture the training dynamics and reduce the computational cost. It bridges the gap between gradient-based approaches like IF and unrolling-based methods like SGD-Influence. While IF is computationally efficient, it struggles with underspecification of the training dynamics. Unrolling-based methods address these limitations but face scalability challenges. SOURCE combines the benefits of both approaches by using an IF-like formula to compute approximate unrolling. This makes SOURCE both computationally efficient and suitable for scenarios where IF struggles, such as non-converged models and multi-stage pipelines. Empirically, SOURCE demonstrates superior performance in counterfactual prediction compared to existing data attribution methods.

## E  DETAILED DISCUSSION OF COMPONENT ATTRIBUTION METHODS

Unlike feature and data attribution, component attribution methods analyze the internal mechanisms of models by attributing model behavior to specific architectural components like neurons, layers, or attention heads. These methods similarly fall into three categories. Perturbation-based methods assess component importance by observing changes in model behavior when specific components are modified, resulting in various forms of causal mediation analysis. Gradient-based methods utilize gradients with respect to component activations to approximate the component importance in causal mediation analysis. Linear approximation methods construct linear models that directly approximate how components affect model behavior. Methods from different categories have their own strengths and limitations. We now extend the discussion of some prominent methods mentioned in the main text in more detail and provide discussions of some additional methods.

### E.1  PERTURBATION-BASED COMPONENT ATTRIBUTION

**Various Types of Ablations in Causal Mediation Analysis**  The causal mediation analysis is frequently referred to as activation patching, wherein activations of the specific component from the clean run are patched into the corrupted run to ascertain if those activations are sufficient and necessary to retrieve the desired output. Activation perturbations can consist of zero ablations (Olsson et al., 2022; Geva et al., 2023), mean ablations (Wang et al., 2022), smoothed Gaussian noising Meng et al. (2022), interchange interventions (Geiger et al., 2021), learned ablations (Li & Janson, 2024). In all cases, the dataset used to generate the activations must be chosen to elicit the desired model behavior, with a matching metric that measures the success of the behavior.

**Automated Circuit Discovery (ACDC)**  Similar to subnetwork pruning, ACDC (Conmy et al., 2023) is tries to find a subnetwork that is far sparser than the original graph and recovers good performance on the task. This is done by iterating through the computational graph of the model from outputs to inputs and attempting to remove as many edges between nodes as possible without reducing the model's performance. In this case, performance is measured as the KL-divergence between the full model and the subgraph's predictions. Furthermore, masked or ablated edges are replaced with activations from a corrupted run or counterfactual input prompt, rather than zero-ablated as is done in subnetwork probing.

### E.2  GRADIENT-BASED COMPONENT ATTRIBUTION

**Edge Attribution Patching (EAP)**  (Syed et al., 2023) combines ACDC and Attribution Patching to create EAP, which generates attribution scores for the importance of all edges in the computational graph through normal attribution patching and then sorts those scores to keep only the top $k$ edges in a circuit, thus yielding the circuit corresponding to the task.

### E.3  GENERALIZING THE DEFINITIONS OF COMPONENTS

While initial works explored this form of causal mediation analysis where each neuron was an individual component (Bau et al., 2020; Vig et al., 2020), recent work has moved towards other mediators due to the computational intractability of considering individual neurons in larger models and due to hypotheses of entanglement and polysemanticity of neurons in foundation models. Furthermore, recent work has argued that specific mediators are only reasonable for certain behaviors

(Mueller et al., 2024) and have also explored the feasibility of patching activations both within and between models to increase expressivity (Ghandeharioun et al., 2024).

**Sparse Autoencoders (SAEs)** (Bricken et al., 2023; Cunningham et al., 2023) are trained to reconstruct model activations under sparsity constraints. Through learning sparse, overcomplete representations of model activations, SAEs effectively decompose complex, entangled features into more interpretable components. The enforced sparsity ensures that each SAE feature captures a distinct and meaningful aspect of the model's behavior, making them useful for model understanding. Recent research has demonstrated that SAEs can successfully extract interpretable components, but since SAEs focus on learning new components rather than attributing to existing ones in the original model, we do not consider them as strictly component attribution methods in this paper. They can rather serve as a technique for discovering interpretable features that can subsequently be used for attribution. As the next paragraph shows, SAEs can be used for component attribution by first discovering interpretable components and then using them for attribution.

**Sparse Feature Circuits** (Marks et al., 2024) Sparse feature circuits build upon the gradient-based attribution method attribution patching to determine the linear directions relevant to the task or behavior of interest. This method leverages sparse autoencoders to find directions in the models's latent space that correspond to human-interpretable features. They then employ linear approximations similar to attribution patching, using either input gradients or integrated gradients, to efficiently identify which of the learned sparse autoencoder features are most relevant to the model behaviors, as well as connections between these features.

## F CHALLENGES IN ATTRIBUTION METHODS

### F.1 COMPUTATION CHALLENGES OF ATTRIBUTION METHODS

Computation challenges present substantial barriers that often prevent attribution methods from being applied to large-scale AI models, such as the foundation models with billions of parameters. For perturbation-based methods, the curse of dimensionality makes a comprehensive analysis intractable when the number of required perturbations is large. For example, the full power set perturbation. This holds for all three types of attributions including high-dimensional inputs, large training datasets, and models with numerous components. Game-theoretic methods face particular difficulties, as exact computation of the Shapley value is often prohibitively expensive and requires approximation techniques like Monte Carlo sampling. The computational burden is also severer for data attribution methods, which require model retraining for each perturbation. Gradient-based methods are more practical for large-scale models. However, gradients essentially only provide first-order approximations of model behavior, which are inadequate to capture complex model behaviors and more sophisticated gradient formulations are needed for better attribution results. For instance, TracIn require aggregating gradients across multiple stages, while IF demand computation of second-order Hessian matrices, leading to increased computational overhead. Linear approximation methods also face computational hurdles in achieving high-quality approximations. Model behavior can be complex, requiring numerous data points and model evaluations to establish sufficient data for learning accurate linear models. Furthermore, for all three types, most attribution methods must compute results separately for each new test data point, creating additional computational strain when attribution analysis is needed for large datasets.

### F.2 CONSISTENCY OF ATTRIBUTION METHODS

The consistency problem in attribution methods is a significant concern. This challenge is also prevalent across due to variability introduced in sampling, learning processes with stochastic optimization, and also non-trivial hyperparameters. When attribution involves sampling, such as the Monte Carlo sampling in some perturbation-based methods to avoid the full power set perturbation, the inherent randomness leads to varying attribution results. Besides, when attribution involves learning processes with stochastic optimization, as seen in mask-learning perturbation and linear approximation methods, different learning outcomes yield inconsistent attribution results. Many attribution methods rely on hyperparameters that can lead to different attribution outcomes. These include sampling parameters, such as the number of samples used for computing Shapley values.

They also include optimization hyperparameters for various learning approaches, such as learning rates and number of steps in linear approximation methods. Additionally, approximation hyperparameters are needed for quantities that are computationally challenging to calculate directly, such as dampening factors for inverse Hessian-vector products. Further variability is introduced through fundamental design choices, such as the selection of perturbation type in perturbation-based methods, where options include mean perturbation, zero perturbation, and random perturbation. While these different approaches should theoretically produce similar results based on their underlying principles, in practice they often yield notably different attributions.

### F.3 EVALUATION OF ATTRIBUTION METHODS

Evaluating attribution methods presents significant challenges due to the lack of ground truth and the inherent complexity of modern AI systems. These challenges stem partially from the inconsistency problem, the computational cost and generalizability of some evaluation metrics, and the lack of universal definitions of importance and ground truth. These common evaluation approaches and their limitations are summarized below.

**Counterfactual Evaluation** is a widely used approach that assesses attribution methods by comparing their scores with the actual impact of removing or modifying elements. Common metrics include *fidelity*, which evaluates sufficiency by retaining only elements with high attribution scores while removing those with low scores. Conversely, *inverse fidelity* measures necessity by removing elements with high attribution scores while retaining those with low scores. LOO attribution represents a special case of inverse fidelity. For data attribution specifically, more sophisticated metrics like LDS for data attribution compare attribution rankings with the actual impact of removing training data points, with LDS being a sophisticated case of fidelity. An important implicit metric in counterfactual evaluation is *sparsity* or *minimality*, which measures how few elements are needed to achieve high fidelity. Greater sparsity is desirable as it indicates that fewer elements are required for explanation. While counterfactual evaluation provides concrete validation, it faces two major challenges: The computational cost of generating counterfactuals, particularly for data attribution, can be prohibitive. Additionally, the complex interactions between elements may not be fully captured by individual counterfactual evaluations.

**Task-Specific Evaluation** assesses the practical utility of attribution methods in downstream tasks. For instance, feature attribution can help identify feature changes that can flip model outputs, while data attribution scores can detect mislabeled training examples, and component attribution scores can help identify the most important components that allows for model pruning. Attribution methods can be compared based on the performance on these specific tasks. While this approach provides practical validation, its findings may not generalize effectively across different tasks or domains.

**Human Evaluation** relies on domain experts or users to assess the quality and interpretability of attributions. This approach is especially valuable for validating whether attributions align with human understanding and domain expertise. For example, for feature attributions, the attribution results can be considered if they generate clearer visual saliency maps that align with human intuition. While human evaluation provides valuable real-world validation, it can be both subjective and resource-intensive and can only be treated as the gold standard in certain cases.

The development of more robust and comprehensive evaluation frameworks remains a crucial research direction for advancing all attribution methods.

