# OpenReview forum: "Building Bridges, Not Walls: Advancing Interpretability by Unifying Feature, Data, and Model Component Attribution"
_ICLR.cc/2025/Workshop/BuildingTrust — BuildingTrust_

### Official Review · Reviewer_m3NS · 2025-02-20
**This paper provides an informative survey of techniques for feature, data, and model component attributions, identifying strong similarities between approaches across the three domains. I believe this unification is a necessary and significant contribution to the field of interpretability.**

**Rating:** 8
**Confidence:** 4

**Review:**

### Strengths
- I found the categorization of past works into method clusters, particularly the rows in Table 2, to be highly informative and illuminating. The interpretability literature is vast and complex, but this survey paper makes significant strides in organizing it.
- The discussion allocated to each cluster is excellent, providing enough insights into these works without overwhelming the reader. The appendix provides a more thorough assessment of each of these papers, ideal for a practitioner to quickly learn about the various methods.

### Weaknesses
- I believe the unification work of (Han et al., 2022) is in a very similar spirit to this paper (unification of attribution methods). The treatment of this citation in 3.4 is well handled, but I believe this citation should come earlier, perhaps in the introduction.
- Although this survey is extensive, I would have liked to see a little bit more discussion on interaction attributions, with some important works in this field cited. (e.g. How does This Interaction Affect Me? Interpretable Attribution for Feature Interactions)

### Suggestions
- I would attempt to move Table 2 to the main body of the paper. As a researcher in the field, I found this table to be particularly informative. If vertical spacing is an issue, perhaps the table could be reduced to three rows (Perturbation, Gradient, and Linear Models), where the Methods are indicated in line with the text (ex. Occlusions (Zeiler & Fergus, 2014) [Direct])
- Line 17 of the abstract: *argues* -> argue

---

### Official Review · Reviewer_FJDV · 2025-02-22
**Unifying Feature, Data, and Model Component Attribution**

**Rating:** 4
**Confidence:** 4

**Review:**

The authors present a unifying framework for understanding feature, data, and model component attribution via perturbations, gradients, and linear approximations. While the position presented in the paper is interesting, I think the paper's contribution may not offer significantly new insights relative to the existing literature. The methods discussed -- perturbations, gradients, and linear approximations --  are fundamentally highly-correlated concepts in mathematics, and it is not very clear to me how this framework enhances our understanding of attribution methods.

---

### Official Review · Reviewer_PYEy · 2025-03-01
**Unified view of feature, data, and component attribution methods**

**Rating:** 6
**Confidence:** 2

**Review:**

**Quality**

Pros:
- Proposes a novel, unified view to bridge feature, data, and component attribution.  Valuable perspective because these have been mostly studied in separation in the past.
- Detailed analysis of successful methods across the 3 domains, revealing valuable connections
- Novel formalization of unified attribution problem

Cons:
- Lots of restating known facts, lack new insights

**Clarity**

Pros:
- Clearly written, introduces individual fields well
- Notation summary is great!
- Clear classification of attribution methods

Originality

Pros:
- Identified potentially novel common concepts and challenges across types of attribution and proposes promising research directions
- Would help newcomers understand the the big picture

Cons:
- Did not produce anything inherently novel as it is a survey paper

Significance:
- Potential to advance interpretability research by synthesizing ideas and encouraging cross-aspect knowledge transfer
- Potentially benefit broader research in attribution due to its holistic perspective
- The techniques studied could soon be applicable in practice
- Unsure how much of this is novel/insightful

---

### Decision · Program_Chairs · 2025-03-04

Accept